# Preliminary Study on the Effect of a Night Shift on Blood Pressure and Clock Gene Expression

**DOI:** 10.3390/ijms24119309

**Published:** 2023-05-26

**Authors:** Barbara Toffoli, Federica Tonon, Fabiola Giudici, Tommaso Ferretti, Elena Ghirigato, Matilde Contessa, Morena Francica, Riccardo Candido, Massimo Puato, Andrea Grillo, Bruno Fabris, Stella Bernardi

**Affiliations:** 1Department of Medical Surgical and Health Sciences, University of Trieste, Cattinara Teaching Hospital, Strada di Fiume 447, 34149 Trieste, Italy; btoffoli@units.it (B.T.); effetonon@gmail.com (F.T.); fabiola.giudici@cro.it (F.G.); tomferretti@yahoo.it (T.F.); elenasofia.ghirigato6@gmail.com (E.G.); matilde.contessa@outlook.it (M.C.); morena.francica@asugi.sanita.fvg.it (M.F.); riccardo.candido@units.it (R.C.); andrea.grillo@units.it (A.G.); b.fabris@fmc.units.it (B.F.); 2SC Patologie Diabetiche, ASUGI, 34100 Trieste, Italy; 3SSD Angiologia e Fisiologia Clinica Vascolare Multidisciplinare Cattinara Teaching Hospital, Strada di Fiume 447, 34149 Trieste, Italy; massimo.puato@asugi.sanita.fvg.it; 4UCO Medicina Clinica, ASUGI Azienda Sanitaria Universitaria Giuliano-Isontina, Cattinara Teaching Hospital, Strada di Fiume 447, 34149 Trieste, Italy

**Keywords:** night shift, clock genes, circadian rhythm, blood pressure, ABPM, dipping, physician, internal medicine

## Abstract

Night shift work has been found to be associated with a higher risk of cardiovascular and cerebrovascular disease. One of the underlying mechanisms seems to be that shift work promotes hypertension, but results have been variable. This cross-sectional study was carried out in a group of internists with the aim of performing a paired analysis of 24 h blood pressure in the same physicians working a day shift and then a night shift, and a paired analysis of clock gene expression after a night of rest and a night of work. Each participant wore an ambulatory blood pressure monitor (ABPM) twice. The first time was for a 24 h period that included a 12 h day shift (08.00–20.00) and a night of rest. The second time was for a 30 h period that included a day of rest, a night shift (20.00–08.00), and a subsequent period of rest (08.00–14.00). Subjects underwent fasting blood sampling twice: after the night of rest and after the night shift. Night shift work significantly increased night systolic blood pressure (SBP), night diastolic blood pressure (DBP), and heart rate (HR) and decreased their respective nocturnal decline. Clock gene expression increased after the night shift. There was a direct association between night blood pressure and clock gene expression. Night shifts lead to an increase in blood pressure, non-dipping status, and circadian rhythm misalignment. Blood pressure is associated with clock genes and circadian rhythm misalignement.

## 1. Introduction

Shift work has been found to be associated with a higher risk of cardiovascular and cerebrovascular disease [1,2]. The mechanisms underlying this association include the fact that rotating night shifts seem to promote cardiovascular risk factors, such as hypertension, diabetes, and obesity, although results have been variable [3].

The negative effects of night shifts on health seem to be due to the disruption of the normal 24 h circadian rhythm. This rhythm is based on a master clock that resides in the suprachiasmatic nucleus of the hypothalamus, which is entrained by light, and on peripheral clocks that integrate signals coming from the master clock as well as the periphery, such as light and food [4,5]. The clocks regulating circadian rhythm rely on cellular networks of transcription factors (core clock genes) that control circadian variations in cellular gene expression, which in turn regulate most physiological functions over 24 h, such as respiration, metabolism, endocrine and immune function, body temperature, and blood pressure. Blood pressure circadian rhythm, for example, is characterized by a morning surge at wake, a plateau during the day, and a nocturnal decline of 10–20% as compared to the average daytime blood pressure, which is known as dipping [6,7].

A person working at night and sleeping during the day suffers from a state of desynchrony, being unable to undergo complete adaptation because of the many external stimuli promoting a day-oriented schedule, as evidenced by the lack of complete adaptation of the core body temperature, melatonin levels, and cortisol rhythm [8]. Desynchrony or circadian misalignment, which might occur as a result of shift work or significant activity at night, has been associated with an increase in blood pressure, and non-dipping blood pressure status [9], and it can also contribute to organ dysfunction and development of disease [10].

This observational study was carried out in a group of internists with the aim of performing a paired analysis of 24 h blood pressure in the same physicians working a day shift and then a night shift, and a paired analysis of clock gene expression after a night of rest or a night of work.

## 2. Results

### 2.1. Population Characteristics

The general characteristics of the 25 internists on shift work are reported in Table 1. The median age was 33 years. There were 15 women (60%) and 10 men (40%), and the average body mass index (BMI) was 22.6. Only one physician was obese. Half of them had a sedentary lifestyle (52%). Only three physicians reported suffering from medical conditions (hypertension, asthma, and Hashimoto’s thyroiditis), and five of them were taking medication (olmesartan, on-demand salbutamol inhalation, levothyroxine, and in two cases oral contraceptive pill). Among these physicians, 12 were resident physicians (48%) and 13 were attending physicians (52%). Overall, the average working age was 7 years. During the day shift, physicians took an average of 7720 steps. During the night shift, on average, physicians received 17.5 calls, took care of 8.5 admitted patients, and took 6660 steps.

### 2.2. 24 h Ambulatory Blood Pressure Measurements

Figure 1a,b show 24 h blood pressure and Figure 1c,d show heart rates as assessed by two consecutive ambulatory blood pressure (BP) monitors (ABPMs), worn by all 25 internists. The first ABPM (Figure 1a,c; control = cnt) was worn on a day with a day shift at work (8–20) and a night of rest, and the second one (Figure 1b,d, night shift = ns) was worn on a day with a day off from work, a night shift (20–08), and a subsequent period of rest (recovery time).

The paired analysis of ABPM results showed that day diastolic blood pressure (DBP) and day heart rate (HR) were higher when physicians were at work (Figure 2b,c). Night systolic blood pressure (SBP), night DBP, and night HR increased during the night shift as compared to the night of rest. DBP decreased during the recovery time, while SBP and HR did not. A repeated-measures ANOVA model was performed to compare SBP, DBP, and their respective dippings during the night of rest, the night shift, and the recovery time after the night shift (Table 2). During the night shift, SBP increased by an average of 10.3 mmHg and DBP increased by an average of 12.1 mmHg as compared to the night of rest. During the recovery time, SBP was reduced by an average of 6.1 mmHg, and DBP was reduced by an average of 9.1 mmHg. In addition, the paired analysis of ABPM results showed that the night shift significantly reduced SBP, DBP, and HR dipping (Figure 2d), corresponding to the nocturnal decline in SBP, DBP, and HR [6]. SBP dipping was 12% (10.5; 14%) on a day with a night of rest, while it was 1.4% (−1.4; 3.7%) on a day with a night of work. DBP dipping was 20.5% (18; 24%) on a day with a night of rest, while it was −0.4% (−2.5; 2.2%) on a day with a night of work. HR dipping was 19% (12; 22%) on a day with a night of rest, while it was 7% (−1.9; 9.6%) on a day with a night of work. In the recovery time after the night shift, dipping remained significantly lower as compared to the dipping during a night of rest (Figure 2d). These data were confirmed by the repeated-measures ANOVA model (Table 2), showing that SBP dipping failed to normalize during the recovery time, as in the recovery time it remained significantly lower as compared to the SBP dipping during the night of rest.

### 2.3. Biochemical Parameters and Circadian Rhythm Gene Expression

All subjects underwent blood sampling after fasting twice: after a night of rest and after a night of work. Overall, we found no differences in the levels of glucose, cortisol, melatonin, C-reactive protein (CRP), interleukin (IL)-1β, IL-6, and tumor-necrosis factor (TNF)-α after a night of rest as compared to a night of work (Table 3). On the other hand, after a night shift, we found that there was a significant increase in the expression of most circadian rhythm genes, i.e., *BMAL*, *CLOCK*, *PER1*, *PER2*, and *PER3* (Figure 3a–e), whereas the gene expression of *CRY1*, CRY2, *IL-6*, and *TNF-α* did not change (Figure 3f–i).

### 2.4. Correlation Coefficients and Linear Regression Analysis

Focusing on night blood pressure, night SBP and night DBP were related to sex (Figure 4) and BMI (Table 4), while age and workload, as assessed by the number of calls, admissions, and steps, were not associated with them. On the cnt day, there was a direct correlation between night SBP and *PER3* gene expression on the morning after. On the night shift day, there was a direct correlation between night SBP and night DBP and *PER2* gene expression on the morning after. In addition, *PER2* gene expression on the morning after the night shift was related to the dipping during the recovery period. Otherwise, dippings were not related to age, sex, BMI, or workload.

Multivariate linear regression analysis showed the presence of independent associations between night SBP and *PER3* gene expression on the cnt day and the associations between night SBP and DBP and *PER2* gene expression on the night shift day (Table 5).

### 2.5. Random Forest and Linear Regression Analysis

To further explore the relationship between night blood pressure and clock gene expression, because of the large set of covariates and the small sample size, we used the random forest (RF) as a technique to identify predictors of blood pressure, as shown in the variance importance plots (Figure 5 and Figure 6). The main advantage of selecting relevant variables through this algorithmic modeling technique is that it is independent of any assumptions about the relationships among variables and about the distribution of errors.

Consistent with our previous results, the RF showed that the best predictors of night SBP and DBP during the cnt day were BMI, *PER3*, and sex (Figure 5). The best predictors of night SBP during the night shift day were *PER3*, age, and sex, while the best predictors of night DBP during the night shift day were *PER3*, years of work, and sex (Figure 6).

Multivariate linear regression analysis confirmed that sex, BMI, and *PER3* gene expression were independently associated with night SBP on the cnt day. BMI and *PER3* gene expression were independently associated with night DBP on the cnt day. Sex and *PER3* gene expression were independently associated with SBP and DBP on the night shift day (Table 6).

## 3. Discussion

Here, we show that in a group of internists with a median age of 33 years, the night shift was associated with a significant increase in SBP, DBP, and HR values and a significant reduction in blood pressure and heart rate nocturnal decline (dipping), such that participants had a blood pressure and heart rate non-dipping profile during their night shift. Blood pressure has a circadian rhythm, and, at night, it generally decreases by 10–20%, which is known as dipping. The loss of this blood pressure nocturnal decline is known as non-dipping and it is associated with metabolic and cardiovascular disease [6,11,12,13,14]. In our study, night shifts reduced SBP dipping from 12.6% to 1.49% and DBP dipping to 20.7% to 0.47%. These findings are consistent with the work of Yamasaki et al., who found an association between the night shift and non-dipping blood pressure status [9], which could indicate a mechanism whereby rotating night shifts lead to a higher risk of cardiovascular and cerebrovascular disease [1,2,15].

Our data show that during the night shift, SBP increased by an average of 10.3 mmHg and DBP increased by an average of 12.1 mmHg, which is consistent with previous work demonstrating an association between shift work and blood pressure. In general, shift workers seem to be more likely to have hypertension (OR 1.31, 95% CI 1.07–1.6) [16]. In addition, shift workers have a 2.1-fold-increased odds ratio of new antihypertensive medication use as compared to non-shift workers [17], and average sleep duration is inversely associated with blood pressure [18]. The association between shift work and blood pressure has been documented in healthcare workers as well. For example, in a group of physicians working in an emergency department with an age range from 28 to 40 years, DBP was significantly higher during night shift activity, as compared to non-shift awake or sleep [19]. Among female nurses, there was an additive interaction between age and the number of night shifts per month on hypertension prevalence [20]. In particular, the number of night shifts was associated with the prevalence of hypertension in participants in the age range of 36–45 years [20].

Our study also showed that after a night shift, there was a significant increase in the expression of most clock genes, namely *BMAL*, *CLOCK*, *PER1*, *PER2*, and *PER3*, which is consistent with circadian rhythm misalignment. The human 24 h circadian rhythm is controlled by a master clock in the suprachiasmatic nucleus (SNC) and by peripheral clocks that integrate signals coming from the master clock as well as the periphery, such as light and food [4,5]. These inner clocks rely on cellular transcriptional–translational feedback loops whereby BMAL1 heterodimerizes with CLOCK to promote the transcription of two repressor genes, Period (Per homologs *PER1*, *PER2*, and *PER3*) and Cryptochrome (Cry homologs *CRY1* and *CRY2*). Once they are translated, PER and CRY form a heterodimer that stops the transcription of *BMAL1* and *CLOCK*. Once PER and CRY are degraded, *BMAL1* and *CLOCK* transcription begins again. In normal conditions, PER1, PER2, and PER3 transcripts peak during the biological night, while *BMAL1* transcripts peak during the biological day [21]. It has been shown that sleep deprivation leads to circadian disruption and misalignment with changes in the clock gene expression in human blood cells, such as *PER2* upregulation in a different time period [21]. Additionally, in a mouse model of shift work, two weeks of sleep restriction was associated with changes in core clock gene expression, i.e., an upregulation of *Bmal1* and *Per1* between 0 and 6 am, while overall rhythmicity was preserved [22].

It has been argued that circadian misalignment has an impact on some physiological parameters, such as blood pressure, which is one of the mechanisms whereby shift work can have negative health consequences [3]. In our study, correlation coefficients as well as random forests pointed to an association between circadian rhythm and blood pressure. Although there is a lack of consensus on the appropriate sample size for multiple regression, linear regression analyses confirmed the existence of independent significant associations between night SBP and DBP and clock gene expression, particularly *PER3* expression both on the cnt and night shift day. By contrast, night workload as assessed by steps, the number of calls, and admissions was not related to the blood pressure level, possibly because light exposure (which is experienced during night shifts) may be a sufficient suppressor of circadian rhythm [23,24], which ultimately affects blood pressure. Animal studies have demonstrated an important role for the clock genes—and PER homologs—in the regulation of blood pressure [25]. Deletion of *Per1* in hypertensive male mice resulted in lower blood pressure as compared to wild-type mice, suggesting that loss of *Per1* was protective in the setting of hypertension [26]. Cell culture experiments show that *Per1* is a positive regulator of sodium reabsorption genes [27,28]. Additionally, human studies have shown that *PER1* was upregulated in the kidneys of hypertensive subjects [29]. In addition, it has been shown that clock gene variants explained 7.1% of the variance in SBP, following adjustments for age, sex, BMI, medication, and social characteristics in a non-Hispanic white population [30], and significant associations were found between SBP and *CLOCK*, *PER1*, *PER3*, and *CRY1* [30]. Other studies have shown associations between *CRY1* and *CRY2* polymorphisms and blood pressure in patients with metabolic syndrome [31], or associations between *PER2* and *BMAL1* polymorphisms and the non-dipper phenotype in 372 young hypertensive patients [32]. A genetic association study designed to test the relevance of 19 polymorphisms of *BMAL1* in 1304 individuals showed that two *BMAL1* haplotypes were associated with type 2 diabetes and hypertension [33].

Going back to our study, and its strengths and limitations, its strengths include recruitment of physicians working night shifts in the same setting/hospital and the possibility to perform paired analyses (as the same physicians were tested twice). On the other hand, limitations include the age of physicians (median age was 33), as it is possible that results would differ in older cohorts; the lack of blood sampling and biochemical measurements at night; and the small sample size. The number of physicians recruited did not allow us to assess the impact of other variables on blood pressure in the model for repeated measures, so larger studies are needed to confirm and extend our findings. In particular, we believe that, in future studies, it would be useful to add gene expression and biochemistries at midnight to gain more information on circadian rhythm misalignment and possible hormonal changes during night shifts.

In conclusion, in line with previous work, our study shows that night shift increased blood pressure leading to a non-dipping blood pressure status, which is a cardiovascular disease risk factor. Night shifts increased clock gene expression on the morning after, consistent with circadian misalignment, and there was an independent association between blood pressure during the night shift and *PER2-3* expression. Our data provide further evidence on the negative impact of night work on cardiovascular health as well as on the association between circadian clock misalignment and blood pressure control. These data support interventions aiming to improve sleep and recovery and reduce fatigue, contributing to the sustainable employability of healthcare workers [34].

## 4. Materials and Methods

### 4.1. Study Design and Population

This is an observational cross-sectional study, whose aims were (i) to perform a paired analysis of 24 h blood pressure in the same physicians working a day shift and then a night shift; (ii) to perform a paired analysis of clock gene expression after a night of rest or a night of work.

Participants were consecutively recruited among physicians working in the Department of Internal Medicine (Dipartimento di Medicina of Cattinara Teaching Hospital, ASUGI, Trieste, Italy). Inclusion criteria were aged between 25 and 60 years; working night shifts in the same Department of Internal Medicine; consent to take part in the study. Exclusion criteria were history of cardiovascular or endocrine disorders; children with age <2 years; return to work after holidays in the 3 weeks before enrollment; shifts in COVID-19 wards.

After providing the informed consent, each participant wore an ambulatory BP monitor (ABPM) twice. The first time (control, cnt day) was for a 24 h period (from 09.00 to 09.00) that included a 12 h day shift (08.00–20.00) and a night of rest. The second time (night shift, ns day) was for a 30 h period (from 09.00 to 13.00) that included a day off, a night shift (20.00–08.00), and a period of rest (08.00–13.00) the following day. The ABPM was a SpaceLabs Oscillometric Model 90207 (Redmon, WA, USA). The ABPM was applied by a resident physician, who evaluated and calibrated the readings at time of placement and at removal. Before its placement, we measured anthropometric parameters and collected the history of each participant. Participants were also asked to keep a diary of their activities, write down when they slept on the CNT day, and to provide details of their activity during the night shift. In particular, participants were asked to write down the number of steps, the number of calls they received, and admissions they had to take care of during the night shift. In addition, they were asked to fast from 24.00 to 09.00 on both the cnt and ns days. When the ABPM was removed, participants underwent blood sampling after fasting to measure glucose, cortisol, melatonin, CRP, IL-1β, IL-6, and TNF-α and to evaluate clock gene expression.

Dipping was calculated as follows: (mean wake hours BP − mean sleep BP divided by mean wake hours BP) × 100. Non-dipping status was defined as sleep hours BP or HR dipping of less than 10% [35].

This study was conducted in accordance with the Declaration of Helsinki, and the protocol was approved by the Institutional Review Board and Ethics Committee (CEUR 107_2020H).

### 4.2. Biochemical Parameters and ELISA

Glucose, CRP, and cortisol were measured using an autoanalyzer. Melatonin (E-EL-H2016, Elabscience, Houston, TX, USA), IL-1β (DY201-05, R&D Systems, Minneapolis, MN, USA), IL-6 (DY206-05, R&D Systems), and TNF-α (DY210-05, R&D Systems) were measured using ELISA, according to the manufacturer’s instructions. The intra-assay coefficients of variations were 5.28% for melatonin, 3.51% for IL-1β, 5.37% for IL-6, and 3.72% for TNF-α. The inter-assay coefficients of variations were 5.02% for melatonin, 9.91% for IL-1β, 11.40% for IL-6, and 8.74% for TNF-α, as already described [36,37].

### 4.3. Peripheral Blood Mononuclear Cell Isolation and Gene Expression Analysis

To isolate peripheral blood mononuclear cells (PBMCs), blood samples were collected in EDTA tubes, layered onto the Ficoll-Paque^TM^ Plus (Cytiva Sweden AB, Uppsala, Sweden) solution, and then centrifuged at 2400 rpm for 30 min at room temperature. The mononuclear cell layer that was obtained was used to extract RNA, as already described [38,39].

RNA extraction was performed with the AllPrep DNA/RNA Mini Kit (Qiagen, Hilden, Germany), according to the manufacturer’s instructions. A total of 700 ng of RNA was subsequently used to synthesize cDNA with the Superscript First-Strand Synthesis System for RT-PCR (Gibco BRL, Thermo Fisher Scientific, Waltham, MA, USA). The expression of genes related to circadian rhythm (*BMAL*, *CLOCK*, *PER1*, *PER2*, *PER3*, *CRY1*, *CRY2*) and inflammation (*IL-6* and *TNF-α*) was analyzed via RT-qPCR using the SYBR Green system (Thermo Fisher Scientific). Primer sequences used with the SYBR Green system are reported in Table 7. Fluorescence for each cycle was quantitatively analyzed using the StepOnePlus Real-Time PCR System (Applied Biosystems, Thermo Fisher Scientific). The PCR was started with a holding stage at 50 °C for 2 min and 95 °C for 2 min, followed by 40 cycles at 95 °C for 3 s and 60 °C for 30 s. A final dissociation step at 95 °C for 15 s, 60 °C for 1 min, and 95 °C for 15 s completed the program. Gene expression was normalized to *GAPDH* and reported as a ratio compared with the level of expression in controls, which were given an arbitrary value of 1.

### 4.4. Statistical Analyses

Sample size was calculated with openepi.com (https://www.openepi.com/SampleSize/SSMean.htm, accessed on 12 April 2023). To detect a mean difference in blood pressure of 4.6 mmHg [19], with a two-sided significance level of 5% and power of 80%, a paired *t*-test would require 19 subjects. Based on this estimate, we decided to include all the internists working in the department who consented to participate in this study. All statistical analyses were carried out in R system for statistical computing (Version 4.0.2; R development Core Team, 2020, Vienna, Austria). Statistical significance was set at *p* < 0.05. A Shapiro–Wilk test was applied to continuous variables to check for distribution normality. Quantitative variables were reported as median with range (1st Qu–3rd Qu) or mean ± standard deviation, depending on distribution. Categorical variables were reported as absolute frequencies and/or percentages. Continuous variables were compared using a Mann–Whitney test (and Kruskal–Wallis test) or Student’s *t*-test (and ANOVA), depending on data distribution and number of groups. A repeated-measures ANOVA model was performed to compare the effect of time on SBP, DBP, SBP dipping, and DBP dipping during the night of rest (t1), the night shift (t2), and the recovery time after the night shift (t3). For every ANOVA model, normality checks were carried out on the dependent variables by group (three different time points), which were approximately normally distributed. If Mauchly’s test of sphericity was not met (only for the dipping DBP model) Greenhouse–Geisser sphericity correction was applied. Post hoc analyses were performed with a Bonferroni adjustment. Correlation coefficients were calculated based on data distribution. Associations with a *p*-value below 0.10 were considered for multivariate linear regression to evaluate factors influencing blood pressure. In addition, in order to identify covariates related to blood pressure, we used random forest regression (RF), which is used when there are several covariates and small sample sizes to select a smaller number of relevant predictors. We retained the top 3 predictors based on mean square error (MSE) decrease in accuracy for each feature removal (%IncMSE), and we applied a linear multivariate regression model to estimate the β-regression coefficient of each predictor. Statistical analyses were performed with the randomForest package in R.

## Figures and Tables

**Figure 1 ijms-24-09309-f001:**
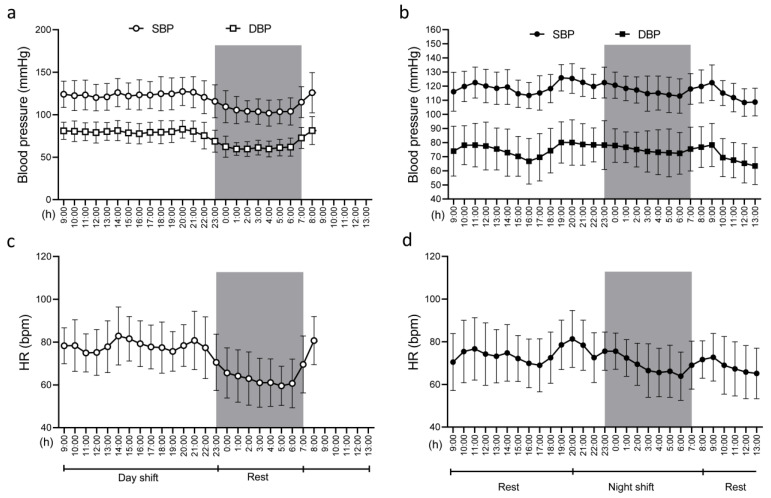
Participants’ 24 h blood pressure and heart rate (HR). (**a**) Blood pressure during control (cnt) day; (**b**) blood pressure during night shift (ns) day; (**c**) heart rate during cnt day; (**d**) heart rate during ns day. Gray areas correspond to the sleep hours on cnt day. Data are presented as mean ± SD.

**Figure 2 ijms-24-09309-f002:**
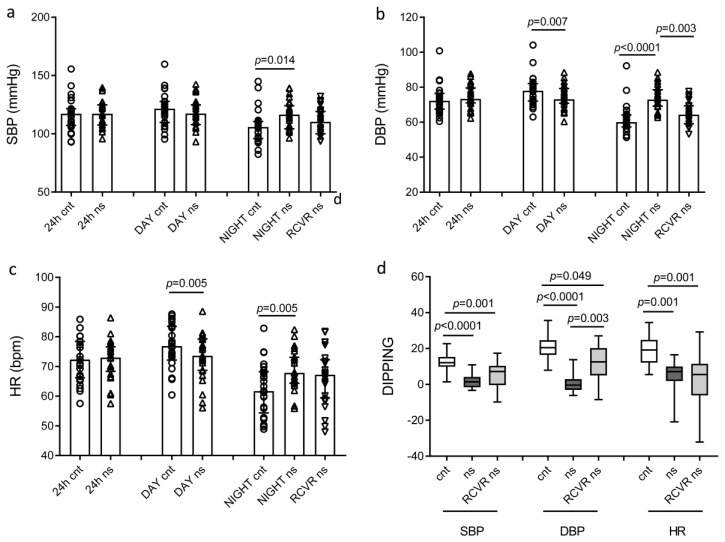
Paired analysis of SBP, DBP, HR, and dippings. (**a**) SBP; (**b**) DBP; (**c**) HR (**d**) dipping. Control (cnt) represents 24 h ABPM worn during the day with a day at work and a night of rest; night shift (ns) represents 24 h ABPM worn during the day with a day off and a night shift; RCVR represents the recovery time after the night shift. Data are presented as median (top of the histogram) with interquartile ranges in (**a**–**c**) and standard box plots in (**d**).

**Figure 3 ijms-24-09309-f003:**
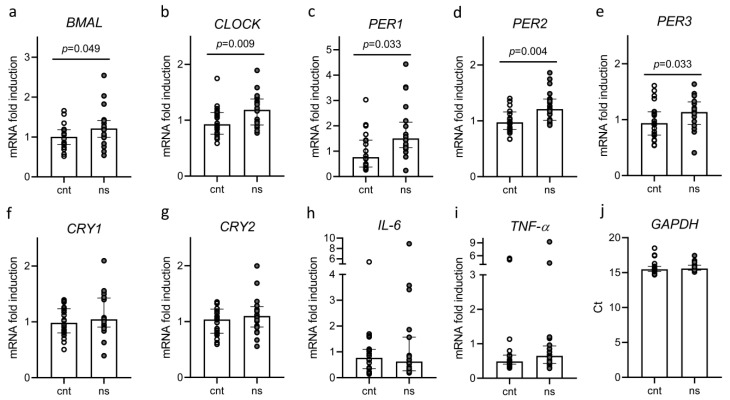
Clock gene expression after a night of rest and after a night shift. Paired analysis of gene expression after a night of rest (cnt) vs. after a night shift (ns). (**a**) *BMAL*; (**b**) *CLOCK*; (**c**) *PER1*; (**d**) *PER2*; (**e**) *PER3*; (**f**) *CRY1*; (**g**) *CRY2*; (**h**) *IL-6*; (**i**) *TNF-α*; (**j**) *GAPDH*. Data are presented as median (top of the histogram) with interquartile ranges. In addition, each individual value is plotted as a point superimposed on the graph.

**Figure 4 ijms-24-09309-f004:**
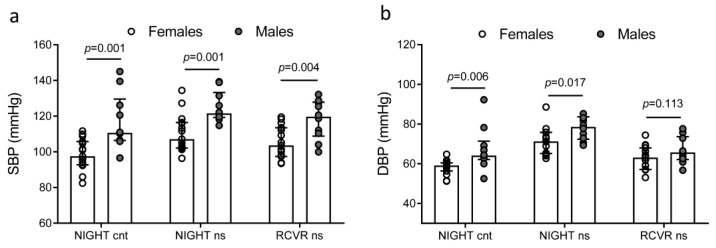
SBP and DBP differences between females and males: (**a**) SBP, systolic blood pressure; (**b**) DBP, diastolic blood pressure. Cnt represents the control day (day with a night of rest), ns represents the night shift day (day with the night shift), RCVR represents the recovery period after the night shift. Data are presented as median (top of the histogram) with interquartile ranges. In addition, each individual value is plotted as a point superimposed on the graph.

**Figure 5 ijms-24-09309-f005:**
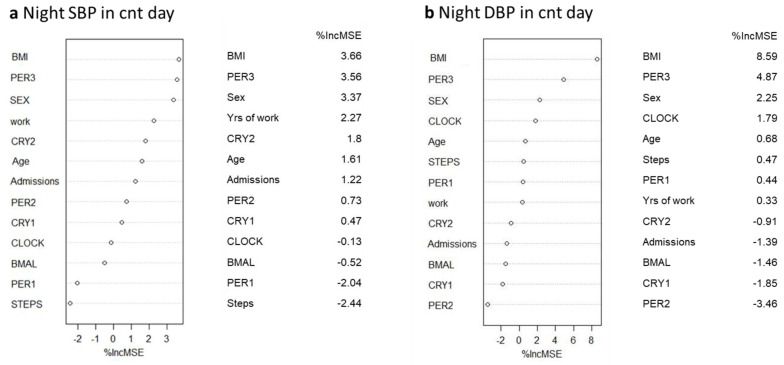
Variance importance plot: %IncMSE represents mean square error (MSE) decrease in accuracy for each feature removal. (**a**) Night SBP, (**b**) Night DBP in a control day.

**Figure 6 ijms-24-09309-f006:**
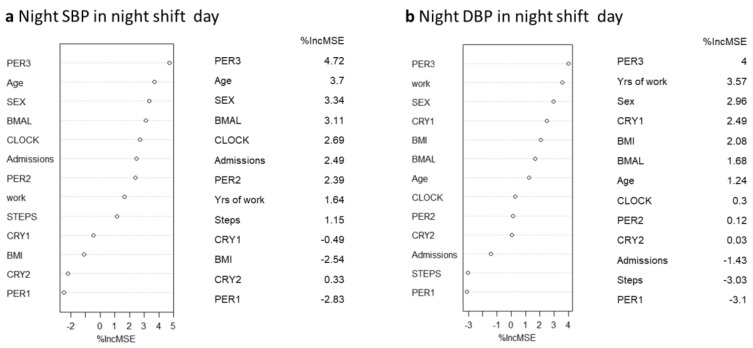
Variance importance plot: %IncMSE represents mean square error (MSE) decrease in accuracy for each feature removal. (**a**) Night SBP, (**b**) Night DBP in a night shift day.

**Table 1 ijms-24-09309-t001:** Population characteristics.

Variable		*n* = 25
Age		33 (29–37)
Sex	Female (%)	15 (60%)
	Male (%)	10 (40%)
BMI		22.6 (20.8–24.54)
Physical activity (Recreational activity)	Yes (%)	12 (48%)
	No (%)	13 (52%)
Medical conditions	Yes (%)	3 (12%)
	No (%)	22 (88%)
Medication	Yes (%)	5 (20%)
	No (%)	20 (80%)
Years of work		7 (2–10)
Working position	Attending physician (%)	13 (52%)
	Resident physician (%)	12 (48%)
Day shift steps		7720 (6504–11,062)
Night shift calls		17.5 (13–20)
Night shift admissions		8.5 (6–10)
Night shift steps		6660 (5785–7822)

Data are expressed as median (1st Qu–3rd-Qu).

**Table 2 ijms-24-09309-t002:** Comparison of night SBP, DBP, and dipping with ANOVA.

	NIGHT cnt (t1)	NIGHT ns (t2)	RCVR ns (t3)	ANOVA	t1 vs. t2	t2 vs. t3	t1 vs. t3
**SBP** **(mmHg)**	106.08 ± 14.82	116.38 ± 12.15	110.24 ± 11.66	*F* = 10.98 *p* < 0.001 η_g_^2^ = 0.1	*p* < 0.001	*p* = 0.006	*p* = 0.28
**DBP** **(mmHg)**	61.85 ± 8.57	73.96 ± 7.12	64.87 ± 6.64	*F* = 30.72 *p* < 0.001 η_g_^2^ = 0.33	*p* < 0.001	*p* < 0.001	*p* = 0.23
**SBP** **dipping (%)**	12.6 ± 4.82	1.49 ± 3.69	5.01 ± 7.51	*F* = 27.56 *p* < 0.001 η_g_^2^ = 0.42	*p* < 0.001	*p* = 0.17	*p* < 0.001
**DBP** **dipping (%)**	20.7 ± 6.58	0.47 ± 4.67	11.7 ± 10.6	*F* = 42.42 *p* < 0.001 η_g_^2^ = 0.55	*p* < 0.001	*p* < 0.001	*p* < 0.001

“NIGHT cnt” represents the night of rest during the control day (t1); “NIGHT ns” represents the night during the night shift (t2); “RCVR ns” represents the recovery time after the night shift (t3). Data are presented as mean ± SD.

**Table 3 ijms-24-09309-t003:** Biochemical parameters after a night of rest and after a night of work.

Variable	After Night of Rest	After a Night Shift	*p*-Value
Glucose (mg/dL)	88.5 (80–91)	86 (80–96)	*p* = 0.35
Cortisol (nmol/L)	303 (260.5–362)	331 (269–413)	*p* = 0.11
Melatonin (pg/mL)	108.1 (98–158)	127.8 (99–138)	*p* = 0.58
CRP (mg/L)	0.85 (0.4–1.5)	0.7 (0.6–1.4)	*p* = 0.11
IL-1β (pg/mL)	12.08 (0–69)	12.89 (0–72)	*p* = 0.41
IL-6 (pg/mL)	28.81 (3–117.5)	30.77 (2–156)	*p* = 0.15
TNF-α (pg/mL)	5.13 (0–82.5)	3.20 (0–76)	*p* = 1.00

Data are expressed as median (1st Qu–3rd Qu).

**Table 4 ijms-24-09309-t004:** Correlation coefficients of night SBP and DBP values.

Variable	Cnt Day				Night Shift Day			
	Night SBP		Night DBP		Night SBP		Night DBP	
	rho	*p*	rho	*p*	rho	*p*	rho	*p*
**Age**	0.01	0.96	0.10	0.65	−0.23	0.28	0.18	0.37
**BMI**	0.42	0.04 *	0.49	0.01 *	0.40	0.05 *	0.40	0.045 *
** *BMAL* **	0.18	0.43	0.09	0.70	−0.22	0.35	−0.09	0.69
** *CLOCK* **	0.30	0.20	0.05	0.83	0.36	0.12	0.33	0.15
** *PER1* **	−0.02	0.94	−0.09	0.71	−0.17	0.46	−0.05	0.82
** *PER2* **	0.23	0.33	0.04	0.87	0.40	0.08 *	0.45	0.04 *
** *PER3* **	0.51	0.02 *	0.27	0.25	0.37	0.11	0.35	0.13
** *CRY1* **	0.22	0.35	0.09	0.71	0.13	0.58	0.17	0.47
** *CRY2* **	0.32	0.17	0.30	0.20	0.23	0.34	0.29	0.22
**Night calls**	NA	NA	NA	NA	0.15	0.63	0.20	0.54
**Night admissions**	NA	NA	NA	NA	0.01	0.96	0.07	0.74
**Night steps**	NA	NA	NA	NA	−0.10	0.66	−0.01	0.96

* Statistically significant variables at a *p*-value of <0.10 level were selected for multivariate linear regression. NA represents not applicable.

**Table 5 ijms-24-09309-t005:** Multivariate linear regression.

**Dependent Variable: Night SBP (cnt Day)**				
**Predictive Variable**	**β-Estimate**	**SE**	***p*-Value**	**Multiple R-Squared**
**Sex**	9.33	3.96	0.03	
**BMI**	1.69	0.60	0.01	0.72
** *PER3* **	18.84	5.36	0.003	
**Dependent Variable: Night SBP (Night Shift Day)**				
**Predictive Variable**	**β-Estimate**	**SE**	***p*-Value**	**Multiple R-Squared**
**Sex**	10.52	5.15	0.06	
**BMI**	0.85	0.77	0.28	0.41
** *PER2* **	17.48	8.06	0.045	
**Dependent Variable: Night DBP (Night Shift Day)**				
**Predictive Variable**	**β-Estimate**	**SE**	***p*-Value**	**Multiple R-Squared**
**Sex**	4.34	3.19	0.19	
**BMI**	0.50	0.48	0.31	0.48
** *PER2* **	11.70	4.99	0.03	

**Table 6 ijms-24-09309-t006:** Multivariate linear regression after random forest selection of predictive variables.

**Dependent Variable: Night SBP (cnt Day)**				
**Predictive Variable**	**β-Estimate**	**SE**	***p*-Value**	**Multiple R-Squared**
**Sex**	−10.36	3.38	0.007	
**BMI**	1.74	0.52	0.004	0.76
** *PER3* **	19.08	4.93	0.001	
**Dependent Variable: Night DBP (cnt Day)**				
**Predictive Variable**	**β-Estimate**	**SE**	***p*-Value**	**Multiple R-Squared**
**Sex**	−3.5	2.12	0.12	
**BMI**	0.99	0.33	0.008	0.61
** *PER3* **	6.79	3.10	0.04	
**Dependent Variable: Night SBP (Night Shift Day)**				
**Predictive Variable**	**β-Estimate**	**SE**	***p*-Value**	**Multiple R-Squared**
**Age**	−0.11	0.35	0.75	
**Sex**	−18.37	4.11	<0.001	0.63
** *PER3* **	25.34	7.23	0.003	
**Dependent Variable: Night DBP (Night Shift Day)**				
**Predictive Variable**	**β-Estimate**	**SE**	***p*-Value**	**Multiple R-Squared**
**Sex**	−8.50	2.81	0.008	
**Years of work**	−0.23	0.25	0.37	0.51
** *PER3* **	12.38	4.99	0.02	

**Table 7 ijms-24-09309-t007:** Primer sequences.

Target (GenBank Accession Number)	Primer Pair	Mature Transcript	Amplicon Size (bp)
*BMAL1* (NM_001297719)	(F) 5′-TTACTGTGCTAAGGATGGCTG-3′ (R) 5′-GCCCTGAGAATGAGGTGTTTC-3′	(F) 731–751 (R) 857–837	127
*CLOCK* (NM_004898)	(F) 5’-CTACATTCACTCAGGACAGGC-3’ (R) 5’-GGCCCATAAGCATAGTACTAGG-3’	(F) 2495–2515 (R) 2614–2593	120
*PER1* (NM_002616)	(F) 5′-CCTCCAGTGATAGCAACGG-3′ (R) 5′-ACCAGATGCACATCCTTACAG-3′	(F) 1784–1802 (R) 1874–1854	91
*PER2* (NM_022817)	(F) 5′-GCCAGAGTCCAGATACCTTTAG-3′ (R) 5′-TGTGTCCACTTTCGAAGACTG-3′	(F) 505–526 (R) 602–582	98
*PER3* (NM_001377275)	(F) 5′-CTGTCTCACTGTGGTTGAAAAG-3′ (R) 5′-TAGGTAACCCAGCAAAGGC-3′	(F) 1063–1084 (R) 1207–1189	145
*CRY1* (NM_004075)	(F) 5’-TCCCGTCTGTTTGTGATTCG-3’ (R) 5’-TTAATAGCTGCGTCTCGTTCC-3’	(F) 799–818 (R) 929–909	131
*CRY2* (NM_021117)	(F) 5′-TGGATAAGCACTTGGAACGG-3′ (R) 5′-AGAGACAACCAAAGCGCAG-3′	(F) 735–754 (R) 854–836	120
*TNF-α* (NM_000594)	(F) 5′-ACTTTGGAGTGATCGGCC-3′ (R) 5′-GCTTGAGGGTTTGCTACAAC-3′	(F) 332–349 (R) 470–451	139
*IL-6* (NM_000600)	(F) 5′-CCACTCACCTCTTCAGAACG-3′ (R) 5′-CATCTTTGGAAGGTTCAGGTTG-3′	(F) 199–218 (R) 348–327	150
*GAPDH* (NM_002046)	(F) 5’-CATCCATGACAACTTTGGTATCGT-3’ (R) 5’-CCATCACGCCACAGTTTCC-3’	(F) 565–588 (R) 672–654	108

## Data Availability

The raw data supporting the conclusions of this article will be made available by the authors upon reasonable request.

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
