# Peer review of "Preliminary Study on the Effect of a Night Shift on Blood Pressure and Clock Gene Expression"

_ijms, 2023, doi:10.3390/ijms24119309_

Round 1

Reviewer 1 Report

Thank You for the opportunity to review this manuscript. Please, find my remarks below.

Minor remarks:

- Table 4: Could the Authors show the exact sites in which the primers are bound to the target RNA?

- lines 279 - 281: 'Quantitative variables were reported as median with range (min-max) or mean ± standard deviation, depending on distribution.'. Have the Authors tried to transform the variables for which the distribution was not normal?

Major remarks:

- Section 4.3.: It is written that gene expression was standardised with use of the housekeeping gene - GAPDH. May I invite the Authors to show its expression in the manuscript so as to enable the potential Readers to make sure that GAPDH expression was not affected by the studied effect (night shift)?

- Although I approve of the idea to a priori check the statistical sample size before performing the study, the chosen methology is highly flawed. From what I read in the manuscript, the same subjects were undergoing day and night shifts. Thus, the design and statistical analysis should be different. Multi-way repeated measures ANOVA should be applied, after checking the assumptions (such as sphericity) and performing their related corrections, if needed. Time should be the main effect. Its interactions with the shift, sex and other possible factors which may have an effect on sleep cycle (such as cortisol) should be added to the model. Residuals of the model should be checked for credibility. Such a model would resemble real life interactions, not just the 'shift' effect itself...  but then, it would need a bigger sample size. In its current form, just correlations are reported, which is not the best way to handle such data. All the factors which may have possibly affected the studied signal are raveled and cannot be assessed.

- The population sample is inadequately described. What factors other than steps etc. were controlled? Were some factors like possible depression or obesity considered? Could something have biased the results?

- Table 1: The Authors provided some information regarding the daily schedule of the test subjects: calls, admissions, steps etc. Giving such information there proves confusing to me since these variables were not used as covariates of any sort. Therefore, why were they measured? We could see, looking at the min-max ranges, that these values greatly varied between the test subjects. It could have affected the study.

- line 119: 'Data not shown' - why is it not shown?

- line 101 - 102: 'Overall, we found no differences in the levels of glucose, cortisol, melatonin, CRP, IL-1β, IL-6, TNF-α after a night of rest as compared to a night of work.'. I could not find this data in the manuscript. Thus, I am unable to check this assumption. This affects the credibility of the study.

- line 285 - 286: 'Associations with p value below 0.10 were considered for multivariate linear regression to evaluate factors influencing blood pressure.'. Did the Authors check the assumption of multiple regression? Moreover, the sample size is too low. There should be about 20 - 25 records per variable in this model.

- Table 4: The PCR conditions should be given (temperature and times of each phase of the reaction. Moveover, the size (bp) of the amplicons should be given.

- Boxplots: Box and whiskers should be described. Are whiskers denoted by Tukey's W = 1.5 or some other value?

To sum it all up, the design of the study and the choice of methods used to control its bias and analyze the possible effects are extremely basic. I believe that they do not fit the current standards in performing factorial analysis. Therefore, I vote for rejecting the manuscript.

Author Response

We thank Reviewer 1 for their review and valuable comments, which gave us the opportunity to improve our manuscript. Please find the point-by-point reply to all the comments that have been raised below. Changes are in red ink throughout the manuscript. 

Reviewer 1

- Table 4: Could the Authors show the exact sites in which the primers are bound to the target RNA?

Reply: taking into account the Reviewer comment, we have now modified Table 6 on page 10 to provide this information (based on genebank accession number and mature transcript position). The specificity of the primers has been checked with the software Primer-BLAST (http://www.ncbi.nlm.nih.gov/tools/primer-blast)

- lines 279 - 281: 'Quantitative variables were reported as median with range (min-max) or mean ± standard deviation, depending on distribution.'. Have the Authors tried to transform the variables for which the distribution was not normal?

Reply: with respect to the variables that were not normally distributed, we preferred analyzing the 'original' data and performing the corresponding non-parametric statistical analysis. Even if log-transformation or other methods are widely used in biomedical research to deal with skewed data, the results of parametric statistical tests applied to data transformations could be not relevant for the original data and - most importantly - not easy to interpret.

- Section 4.3.: It is written that gene expression was standardised with use of the housekeeping gene - GAPDH. May I invite the Authors to show its expression in the manuscript so as to enable the potential Readers to make sure that GAPDH expression was not affected by the studied effect (night shift)?

Reply: taking into account the Reviewer suggestion, we have now added GAPDH expression to Figure 3, on page 5.

- Although I approve of the idea to a priori check the statistical sample size before performing the study, the chosen methology is highly flawed. From what I read in the manuscript, the same subjects were undergoing day and night shifts. Thus, the design and statistical analysis should be different. Multi-way repeated measures ANOVA should be applied, after checking the assumptions (such as sphericity) and performing their related corrections, if needed. Time should be the main effect. Its interactions with the shift, sex and other possible factors which may have an effect on sleep cycle (such as cortisol) should be added to the model. Residuals of the model should be checked for credibility. Such a model would resemble real life interactions, not just the effect itself...  but then, it would need a bigger sample size. In its current form, just correlations are reported, which is not the best way to handle such data. All the factors which may have possibly affected the studied signal are raveled and cannot be assessed.

Reply: the choice to use of ‘basic’ statistical analyses was related to the number of subjects enrolled, because in case of small sample sizes, models with interactions may be difficult to interpret and not powered to detect clinically significant results.

Nevertheless, to address the Reviewer suggestion, we performed a one-way repeated measures ANOVA model to compare blood pressure at the time points of interest (i.e. t1= night of rest; t2= night shift; t3=recovery time after night shift), whose results are reported on page 3 lines 94-98 and page 4 lines 109-111 as well as on Table 2. These results confirm our previous results. In the section “statistical analyses” (page 11, lines 322-329) we added the following sentence: “A repeated measures ANOVA model was performed to compare the effect of time on SBP, DBP, dipping SBP and dipping DBP during the night of rest (t1), the night shift (t2), and the recovery time after the night shift (t3). For every ANOVA model, normality checks were carried out on the dependent variables by group (three different time points), which were approximately normally distributed. If the Mauchly’s test of sphericity was not met (only for the dipping DBP model) the Greenhouse-Geisser sphericity correction was applied. Post-hoc analyses were performed with a Bonferroni adjustment”.

In addition, in the Discussion, we added a paragraph with strengths and limitations of the study, where we included the issue of the small sample that did not allow us to assess the impact of other variables in the model for repeated measures (page 8, lines 230-232).

- The population sample is inadequately described. What factors other than steps etc. were controlled? Were some factors like possible depression or obesity considered? Could something have biased the results?

Reply: we added further information on the physicians recruited, which include the lifestyle of physicians (regular physical activity), their medical conditions (if any), medication, as well as the number of steps during the day shift. These data are reported on page 2, lines 67-72, and on Table 1. None of them suffered from depression, and only 1 physican was obese. 

- Table 1: The Authors provided some information regarding the daily schedule of the test subjects: calls, admissions, steps etc. Giving such information there proves confusing to me since these variables were not used as covariates of any sort. Therefore, why were they measured? We could see, looking at the min-max ranges, that these values greatly varied between the test subjects. It could have affected the study.

Reply: we actually checked if there was an association between night blood pressure and workload (as assessed by calls, admissions, and steps) but we did not find any. In the first version of the manuscript we wrote: “Focusing on night blood pressure, night SBP and DBP in both control (cnt) and night shift (ns) days were related to sex (Figure 4) and BMI (Table 2), while age or workload as assessed by the number of calls, admissions and steps were not associated with them”. We left this sentence on page 6, lines 139-141.

- line 119: 'Data not shown' - why is it not shown?

Reply: we decided not to show the data on the associations between dippings, age, sex, BMI, and workload because, based on our analysis dippings were not related to age, sex, BMI, and workload.

- line 101 - 102: 'Overall, we found no differences in the levels of glucose, cortisol, melatonin, CRP, IL-1β, IL-6, TNF-α after a night of rest as compared to a night of work.'. I could not find this data in the manuscript. Thus, I am unable to check this assumption. This affects the credibility of the study.

Reply: please find the data on Table 3, page 5.

- line 285 - 286: 'Associations with p value below 0.10 were considered for multivariate linear regression to evaluate factors influencing blood pressure.'. Did the Authors check the assumption of multiple regression? Moreover, the sample size is too low. There should be about 20 - 25 records per variable in this model.

Reply: we confirm that we checked the main assumptions underlying multiple regression models (i.e. linearity, independence of covariates, homoskedasticity). As regard as the question of the number of covariates considered in the multiple linear regression model, there is no general consensus. Usually, based on a rule of thumb, the ratio of observations per variable ought to be in the size of order 10, but other Authors recommend 15, others 20, and others state that even 5 or 2 is sufficient. Taking into account the small sample size of our study and at the same time the number of variables considered, we decided first to investigate the presence of univariate correlations (Table 4) and then to select covariates based both on the results of univariate correlations and on the research question. In the Table 5 reporting the multivariate linear regression results, we have now added the Multiple R-Squared which gives information about the good of fitness of the multivariate model:  while the model with Night SBP (cnt day) as dependent variable is quite robust (Multiple R-squared 0.72) the other two are poor (Multiple R-squared < 0.5).  In the discussion, we have underlined that the small sample size is a limitation and larger studies are needed to confirm these relationship.

- Table 4: The PCR conditions should be given (temperature and times of each phase of the reaction. Moveover, the size (bp) of the amplicons should be given.

Reply: Taking into account the Reviewer observation, we have now detailed the PCR conditions, on page 10, lines 304-306 . In particular, the PCR was started with a holding stage at 50°C for 2 min and 95 °C for 2 min, fol-lowed by 40 cycles at 95 °C for 3 s and 60 °C for 30 s. A final dissociation step at 95 °C for 15 s, 60 °C for 1 min and 95 °C for 15 s completed the program. In addition, the size (bp) of the amplicons in now reported on Table 6.

- Boxplots: Box and whiskers should be described. Are whiskers denoted by Tukey's W = 1.5 or some other value?

Reply: In box plots (Figure 2d), as per standard box plot, the line in the middle of the box is the median, the box extends from the 25th to 75th percentiles, and min and max are whiskers. Otherwise, in Figure 2 (a-c), Figure 3 and Figure 4 the top of histograms is the median and whiskers are interquartile ranges.

Reviewer 2 Report

Please provide the information as follows:

1. The clear info about the use /no use antihypertensive drugs among the study group, present and past.

2. The information about use the antidepressants among the study group

3. The information about regular sport activity among the participants of study group

4. It seem to be needed to show the body temperature  related ,for exemple, to control , ns , and rcvr

Author Response

We thank Reviewer 2 for their review and valuable comments, which have helped us to improve the quality of our manuscript. Please find the point-by-point reply to all the comments that have been raised below. Changes are in red ink throughout the manuscript. 

Reviewer 2

  1. The clear info about the use /no use antihypertensive drugs among the study group, present and past.

Reply: we have added the information on medication on page 2, lines 67-70, and on Table 1. Only 1 patient was taking anti-hypertensive drugs (olmesartan).

  1. The information about use the antidepressants among the study group

Reply: we have added the information on medication on page 2, lines 67-70, and on Table 1. None of the study participants was taking antidepressants.

  1. The information about regular sport activity among the participants of study group

Reply: we have added the information on regular sport activity on page 2, lines 67, and on Table 1. Half of the study participants had a sedentary lifestyle (52%). In addition, we have also added the number of steps made during the day shifts.

  1. It seem to be needed to show the body temperature  related ,for exemple, to control , ns , and rcvr.

Reply: unfortunately in the data that we collected there was no body temperature, so we could not add this data.

Reviewer 3 Report

Toffoli et al aimed to investigate the impact of night shift work on blood pressure and clock gene expression in a group of internists. Previous studies have demonstrated an association between night shift work and non-dipping blood pressure status, higher DBP during night shift activity, and an increased odds ratio of new antihypertensive medication use among shift workers. They recruited 25 internists on shift work to participate in their studies. The participants wore an ambulatory blood pressure monitor and underwent fasting blood sampling twice, after a night of rest and after a night shift. The results showed that night shift work increased blood pressure (SBP), diastolic blood pressure (DBP), and heart rate (HR) values and disrupted the circadian rhythm, resulting in a non-dipping blood pressure profile during the night shift. This suggests that rotating night shifts may promote cardiovascular risk factors and increase the risk of cardiovascular and cerebrovascular disease. They also measured the metabolic profile and gene expression in blood samples, finding no difference in metabolites and significant differences in most circadian rhythm genes. I have some comments on this version, which hopefully can help them to improve the quality of the manuscript.

Major issues:

1.     I would recommend using a machine-learning model such as Random Forest or linear regression to predict the change in blood pressure after the night shift based on all available data before the night shift such as age, baseline blood pressure, etc. In this way, it might be possible to quantify the influence of night shift more precisely. The authors can refer to some computational methods developed in the past to predict respiratory disease of infants later in life based on their omics data in their early life (Xuwen Wang, et al, Respiratory Research 2023). Then based on the trained predictive model, they could once again investigate the importance of each feature and compare the feature importance with the correlation results they presented.

Minor comments:

1.     Some acronyms are mentioned before showing the full names. For instance, on line 28, the full names for SBP, DBP, and HR are not shown. Similarly, on line 24, the meaning of BP is not clear. Please show the full name when the acronym is mentioned for the first time.

2.     Line 106: the period is missing at the end of the sentence.

3.     Line 114: “assessed by number of calls” -> “assessed by the number of calls”.

4.     Line 122: “were independent from sex and BMI” -> “were independent of sex and BMI”

5.     Lines 158, 177, and 198: the empty lines can be removed.

No major issues were detected. Some minor comments are included in the suggestions for the authors.

Author Response

We thank Reviewer 3 for their review and valuable comments, which have helped us to improve the quality of our manuscript. Please find the point-by-point reply to all the comments that have been raised below. Changes are in red ink throughout the manuscript. 

Reviewer 3

Major issues:

  1. I would recommend using a machine-learning model such as Random Forest or linear regression to predict the change in blood pressure after the night shift based on all available data before the night shift such as age, baseline blood pressure, etc. In this way, it might be possible to quantify the influence of night shift more precisely. The authors can refer to some computational methods developed in the past to predict respiratory disease of infants later in life based on their omics data in their early life (Xuwen Wang, et al, Respiratory Research 2023). Then based on the trained predictive model, they could once again investigate the importance of each feature and compare the feature importance with the correlation results they presented.

Reply: We thank the Reviewer for the suggestion. We are aware that machine learning is a powerful tool for making predictions from data. The success of a machine learning-based solution and corresponding applications mainly depends on both the data and the learning algorithms. We actually used one of the most popular (and simplest) machine learning modeling techniques i.e. multiple linear regression:  linear regression creates a relationship between the dependent variable (Y) and one or more independent variables (X). We did not applied more sophisticated machine learning methods because, if the data are bad to learn, such as insufficient quantity for training then the machine learning models may become useless or will produce lower accuracy. Moreover, these advanced methods are affected by the lack of interpretability making it difficult to prove relationships within the data. Considering our available data, we believe that the application of “classical” statistical modeling is more adequate to investigate relationships between variables and the significance of those relationships, whilst also catering for prediction.

Minor comments:

  1. Some acronyms are mentioned before showing the full names. For instance, on line 28, the full names for SBP, DBP, and HR are not shown. Similarly, on line 24, the meaning of BP is not clear. Please show the full name when the acronym is mentioned for the first time.

Reply: we have now showed the full names before the indicated acronyms

  1. Line 106: the period is missing at the end of the sentence.
  2. Line 114: “assessed by number of calls” -> “assessed by the number of calls”.
  3. Line 122: “were independent from sex and BMI” -> “were independent of sex and BMI”
  4. Lines 158, 177, and 198: the empty lines can be removed.

Reply: typos have been amended

Round 2

Reviewer 1 Report

I would like to thank the Authors for revising the manuscript. Unfortunately, I have further concerns which could be found below.

- 'Reply: we decided not to show the data on the associations between dippings, age, sex, BMI, and workload because, based on our analysis dippings were not related to age, sex, BMI, and workload.' - May I invite the Authors back this hypothesis up with any evidence from the literature or their own findings?

- In case in which medians are given, I would advise to feature them with 1st and 3rd quartiles instead of the min-max values (they could be shown but are not as essential as the interquartile range). This could give the Readers an idea about the distribution of these values. Moreover, I would like to ask the Authors to include the information on what the whiskers showed in the boxplots (was it min-max or median+/- Tukey's W value of some sort?)

- The 'Discussion' section could be expanded with further indications which factors might have affected the observed results of this study. Remarks on the way the design of the experiment should look like could be provided, given that this study was preliminary.

- Owing to the very small number of the participants, basic methods used for statistical analysis,  a lot of factors not taken into consideration (such as dietary habits, family history etc.) and heterogeneity of the population sample (the Authors admitted that, for example, one participant was obese), I would encourage the Authors to add a 'preliminary study' suffix to the title.

- 'Reply: (...) In the Table 5 reporting the multivariate linear regression results, we have now added the Multiple R-Squared which gives information about the good of fitness of the multivariate model:  while the model with Night SBP (cnt day) as dependent variable is quite robust (Multiple R-squared 0.72) the other two are poor (Multiple R-squared < 0.5).  In the discussion, we have underlined that the small sample size is a limitation and larger studies are needed to confirm these relationship.' - the Authors provided the multiple R-squared values, admitted that two models were inadequate... but this is by no means mentioned anywhere in the manuscript. Moreover, the Discussion should also feature the information on the lack of consensus on the appropriate sample size for multiple regression. 

Overall, the manuscript has been significantly improved (which, in my opinion, still does not make it suitable for this journal). Although this is by no means a meaningful study due to the small sample size and the design of the experiment, the manuscript could prove to provide some ideas to the potential Readers given that the Discussion would be expanded with an expanded information on the possible factors which could have had an influence on the observed effect, and commentary on the proper study design which should be utilized in such a study by potential Readers in the future.

Author Response

We thank Reviewer 1 for their review and valuable comments, which gave us the opportunity to improve our manuscript. Please find the point-by-point reply to all the comments that have been raised below. Changes are in red ink throughout the manuscript. 

Point-by-point reply to Reviewer 1

  1. 'Reply: we decided not to show the data on the associations between dippings, age, sex, BMI, and workload because, based on our analysis dippings were not related to age, sex, BMI, and workload.' - May I invite the Authors back this hypothesis up with any evidence from the literature or their own findings?

Reply: taking into account the Reviewer suggestion, we report below the correlation coefficients showing no association between of age, sex, BMI, circadian rhythm, workload and blood pressure dipping at night.

Variable

Cnt day

Night shift day

Night SBP dipp

Night DBP dipp

Night SBP dipp

Night DBP dipp

rho

p

rho

P

rho

p

rho

p

Age

-0.158

0.449

-0.29

0.1487

0.283

0.3072

0.33

0.11

BMI

-0.298

0.1466

-0.299

0.1453

0.301

0.1434

0.091

0.6638

BMAL

-0.379

0.098

-0.206

0.3831

-0.302

0.1943

-0.203

0.3888

CLOCK

-0.206

0.3814

-0.0389

0.8705

-0.159

0.501

-0.227

0.335

PER1

0.024

0.9215

-0.010

0.967

-0.186

0.4295

-0.127

0.5901

PER2

-0.315

0.1752

-0.071

0.7637

-0.198

0.4015

-0.243

0.3011

PER3

-0.210

0.3723

0.028

-0.243

-0.242

0.3033

-0.190

0.4212

CRY1

0.1268

0.594

0.246

0.2954

-0.2278

0.3339

-0.186

0.4304

CRY2

-0.2116

0.3704

-0.0779

0.7438

0.0423

0.8593

-0.0186

0.9377

Night calls

0.2953

0.3513

-0.0631

0.8454

Night admissions

0.00794

0.9706

-0.0883

0.6814

Night steps

-0.15

 0.5046

-0.1022

0.6499

With respect to sex, which was not associated with blood pressure dippinds, SBP dipping during cnt day did not differ between males and females (mean in group F was 13.92%, mean in group M was 10.57%, p 0.09). DBP dipping  during cnt day did not differ between males and females (mean in group F was 21.98%, mean in group M was 18.75%, p 0.24); SBP dipping during night shift day did not differ between males and females (mean in group F was 1.59%, mean in group M was 1.32%, p 0.85). DBP dipping during night shift day did not differ between males and females (mean in group F was 0.73%, mean in group M was 0.1%, p 0.71).

Due to the lack of significant associations, we still believe that it is not worth showing these results in the manuscript as they would not change or improve our final results and perhaps make the manuscript too verbose. Please note that based on Reviewer 3 requests we have now added further analyses and data to the manuscript (i.e. Figure 5 and Table 6 with random forest and linear regression)

- In case in which medians are given, I would advise to feature them with 1st and 3rd quartiles instead of the min-max values (they could be shown but are not as essential as the interquartile range). This could give the Readers an idea about the distribution of these values. Moreover, I would like to ask the Authors to include the information on what the whiskers showed in the boxplots (was it min-max or median+/- Tukey's W value of some sort?)

Reply: we have changes min-max to 1st and 3rd quartiles in Table 1 and Table 3 and also in the manuscript (page 3, lines 101-107). In figure legends, we clarified that whiskers show interquartile ranges.

- The 'Discussion' section could be expanded with further indications which factors might have affected the observed results of this study. Remarks on the way the design of the experiment should look like could be provided, given that this study was preliminary.

Reply: we have added two references to the impact of light exposure (which is experienced during night shift) on circadian rhythm [23,24], page 10 lines 231-234. Some parts of the discussion have been paraphrased.

- Owing to the very small number of the participants, basic methods used for statistical analysis,  a lot of factors not taken into consideration (such as dietary habits, family history etc.) and heterogeneity of the population sample (the Authors admitted that, for example, one participant was obese), I would encourage the Authors to add a 'preliminary study' suffix to the title.

Reply: we have added the suffix “Preliminary” to the title of our study

- the Authors provided the multiple R-squared values, admitted that two models were inadequate... but this is by no means mentioned anywhere in the manuscript. Moreover, the Discussion should also feature the information on the lack of consensus on the appropriate sample size for multiple regression. 

Reply: we have now stated that there is a lack of consensus on the appropriate sample size for multiple regression in the Discussion on page 10, line 228-229. In addition, we have now carried out a new linear regression analysis, where covariates were selected based on random forests (Figure 5), which shows association between blood pressure and circadian rhythm with better R-square values (Table 6).

Overall, the manuscript has been significantly improved (which, in my opinion, still does not make it suitable for this journal). Although this is by no means a meaningful study due to the small sample size and the design of the experiment, the manuscript could prove to provide some ideas to the potential Readers given that the Discussion would be expanded with an expanded information on the possible factors which could have had an influence on the observed effect, and commentary on the proper study design which should be utilized in such a study by potential Readers in the future.

Reply: we thank the Reviewer for their suggestions. In addition to a bigger sample size, we believe that, in future studies, it would be useful to add gene expression and biochemistries at midnight to gain more information on circadian rhythm misalignement and possible hormonal changes during night shifts (page 10, lines 258-261).

Reviewer 3 Report

I don’t agree with the authors that machine-learning models such as Random Forest are more data-demanding than linear regression. Quite the contrary, I think Random Forest usually generates better performance than linear regression even when the data quality is not very great. The reason that I suggested using machine-learning models such as Random Forest is that such models don’t take the linear assumptions as the linear regression. Therefore, the authors should try those models and then compare whether the inferred important features from those models agree with the results they obtained from linear regression. In this way, it can guarantee the reliability of identified features.

Author Response

We thank Reviewer 3 for their review and valuable comments, which have helped us to improve the quality of our manuscript. Please find the point-by-point reply to all the comments that have been raised below. Changes are in red ink throughout the manuscript. 

Reviewer 3

I don’t agree with the authors that machine-learning models such as Random Forest are more data-demanding than linear regression. Quite the contrary, I think Random Forest usually generates better performance than linear regression even when the data quality is not very great. The reason that I suggested using machine-learning models such as Random Forest is that such models don’t take the linear assumptions as the linear regression. Therefore, the authors should try those models and then compare whether the inferred important features from those models agree with the results they obtained from linear regression. In this way, it can guarantee the reliability of identified features.

Reply: taking into account the Reviewer comment, we have now applied Random Forest analysis as a technique to select the relevant predictors of blood pressure (we integrated paragraph 4.4 Statistical Analysis), and we have performed new multivariare linear regression analyses with the selected variables. The results confirm (with better R-square values) the presence of an association between circadian rhythm and blood pressure (page 7 lines 161-176, Figure 5 and Table 6, and page 13, lines 361-367).

Round 3

Reviewer 1 Report

I would like to thank the Authors for expanding the manuscript. I think that, it its current state, it may be used as a preliminary study and publishing it is acceptable.